# Novel Phenotyping for Acute Heart Failure—Unsupervised Machine Learning-Based Approach

**DOI:** 10.3390/biomedicines10071514

**Published:** 2022-06-27

**Authors:** Szymon Urban, Mikołaj Błaziak, Maksym Jura, Gracjan Iwanek, Agata Zdanowicz, Mateusz Guzik, Artur Borkowski, Piotr Gajewski, Jan Biegus, Agnieszka Siennicka, Maciej Pondel, Petr Berka, Piotr Ponikowski, Robert Zymliński

**Affiliations:** 1Institute of Heart Diseases, Wroclaw Medical University, 50-556 Wroclaw, Poland; blaziak.mikolaj@gmail.com (M.B.); maksym.jura@gmail.com (M.J.); giwanek95@gmail.com (G.I.); agatazdanowicz@gmail.com (A.Z.); mateuszguzik23@gmail.com (M.G.); artur.borkowski.md@gmail.com (A.B.); dr.piotr.gajewski@gmail.com (P.G.); janbiegus@gmail.com (J.B.); piotr.ponikowski@umw.edu.pl (P.P.); robertzymlinski@gmail.com (R.Z.); 2Department of Physiology and Patophysiology, Wroclaw Medical University, 50-368 Wroclaw, Poland; agnieszka.siennicka@umw.edu.pl; 3Institute of Information Systems in Economics, Wroclaw University of Economics and Business, 53-345 Wroclaw, Poland; maciej.pondel@ue.wroc.pl; 4Department of Information and Knowledge Engineering, Prague University of Economics and Business, W. Churchill Sq. 1938/4, 130 67 Prague, Czech Republic; berka@vse.cz

**Keywords:** acute heart failure, machine learning, clustering

## Abstract

Acute heart failure (AHF) is a life-threatening, heterogeneous disease requiring urgent diagnosis and treatment. The clinical severity and medical procedures differ according to a complex interplay between the deterioration cause, underlying cardiac substrate, and comorbidities. This study aimed to analyze the natural phenotypic heterogeneity of the AHF population and evaluate the possibilities offered by clustering (unsupervised machine-learning technique) in a medical data assessment. We evaluated data from 381 AHF patients. Sixty-three clinical and biochemical features were assessed at the admission of the patients and were included in the analysis after the preprocessing. The K-medoids algorithm was implemented to create the clusters, and optimization, based on the Davies-Bouldin index, was used. The clustering was performed while blinded to the outcome. The outcome associations were evaluated using the Kaplan-Meier curves and Cox proportional-hazards regressions. The algorithm distinguished six clusters that differed significantly in 58 variables concerning i.e., etiology, clinical status, comorbidities, laboratory parameters and lifestyle factors. The clusters differed in terms of the one-year mortality (*p* = 0.002). Using the clustering techniques, we extracted six phenotypes from AHF patients with distinct clinical characteristics and outcomes. Our results can be valuable for future trial constructions and customized treatment.

## 1. Introduction

Acute heart failure (AHF) is a life-threatening challenge in a clinical approach, causing a growing number of hospitalizations and a high in-hospital as well as post-discharge mortality range [1]. The present epidemiological situation (e.g., aging population, improved myocardial infarction survival) tends to increase the prevalence of chronic HF, resulting in hospitalization in the near future [2]. Over the years, the approach to the clinical manifestation of AHF has changed; however, it was always crucial for phenotype patients to provide them with a better individual treatment. The AHF diagnostic process starts with the first medical contact and is aimed at identifying the clinical presentation [1]. The clinical severity and medical procedures differ according to a complex interplay between the deterioration cause, underlying cardiac substrate, and comorbidities. It is recommended to stratify AHF patients based on the presence of signs of congestion and/or peripheral hypoperfusion at admission. According to the 2021 ESC Guidelines for the diagnosis and treatment of acute and chronic heart failure, we can distinguish four clinical presentations of AHF: acute decompensated heart failure, acute pulmonary oedema, isolated right ventricular failure and cardiogenic shock, for which phenotyping may bring therapeutic and prognostic value [3]. It could be, nevertheless, questionable if a physical examination and simple dichotomous subgroups sufficiently reflect the complexity of the pathophysiology of AHF and the heterogeneity of AHF patients. These shortcomings in understanding the underlying correlations may be the reason for poor survivability [4].

Machine learning, especially statistical clustering, which is an unsupervised technique that attempts to learn the internal structure of data, might be a feasible tool for elucidating the hidden phenotypic characteristics for a better understanding of the vital differences between clinically important subpopulations [5,6,7,8]. Machine learning approaches have been successfully used in analyzing molecular data for many years. Recently, used with clinical variables, cluster analysis proved itself to be effective in the study of the phenotype characteristics of diseases in chronic heart failure with reduced ejection fraction [9] (HFrEF) as well as a preserved ejection fraction (HFpEF) [5].

According to the aforementioned studies, we implemented machine-learning algorithms for AHF patients and their clinical variables obtained at admission alone. We blinded them to the outcomes to detect novel patterns by subgrouping the patients at the first medical contact. By identifying them in such a manner, we hypothesized that subpopulations of patients would have different pathophysiological characteristics and varying outcomes.

## 2. Materials and Methods

### 2.1. Study Population

We retrospectively analyzed 381 patients hospitalized due to AHF based on two AHF registries that ran at our institution in 2010–2012 and 2016–2017. Patients were treated and heart failure diagnosis was stated following current ESC guidelines. The inclusion and exclusion criteria were elaborated in our previous references [10]. There were no differences in the collected patients’ demographic data or the design of the evaluated registries, except for the criteria of acute heart failure diagnosis, which were slightly varied in the subsequent (2013 and 2016) ESC guidelines.

### 2.2. Machine Learning and Statistical Analysis

As we aimed to evaluate the baseline heterogeneity of AHF patients, only the variables evaluated at admission were included. The analysis was performed blinded to the outcome; therefore, the follow-up variables were excluded (Figure 1). Initially, 88 variables were divided into domains and selected for the study (Table 1). Then, the automatic preprocessing was performed. The low-quality variables were defined as those with over 90% stability and 10% missing values, and 25 such variables were deleted. Furthermore, remove, which correlated with r = 0.6, was implemented, but 0 variables were found and removed. Sixty-three variables were eventually included in the cluster analysis (Table 1). Due to clustering algorithms’ inability to cope with the missing values, they were replaced by mean values. Range transformation normalization (range: 0 to 1) was performed. The nominal parameters were transformed into numerical parameters.

Cluster analysis is an unsupervised machine-learning method which divides the set of variables into smaller groups (clusters) based on their similarity. The clusters are composed of cases which are consistent with each other, but not with other collections. Several clustering algorithms have been described. This analysis uses the k-medoids algorithm to obtain clusters (k-medoids operator in RapidMiner). The number of groups has not been assumed in advance. The optimize parameters operator was used to reveal the most accurate cluster quantity and characteristics. The clustering.k and clustering.numerical_measure parameters were used to optimize the clustering, and the Davies-Bouldin index was chosen as the main criterion. The number of clusters was set between 3 and 6 to avoid excessive dataset fragmentation.

K-medoids is a clustering algorithm that requires that the number of resulting clusters (value of parameter K) is specified in advance. Unlike k-means clustering, where the centroids are computed as the average values of data points (examples) within a cluster, the centroids in the k-medoids algorithm corresponds to the existing data points. This makes the centroids better interpreted. The clustering is based on measuring the distance between the examples; examples in a cluster are similar to each other. The clustering algorithm repeatedly re-assigns the examples into a given number of clusters by minimizing their distance to a centroid and recomputes the centroids. Thus, the concrete distance measure is another important parameter of the method.

Thanks to the option of the automated parameter tuning implemented in RapidMiner, we allowed the system to change the number of clusters K in the range of 3 to 6 and the numeric distance/similarity measure to take any value from the list:EuclideanDistance;CamberraDistance;ChebychevDistance;CorrelationSimilarity;CosineSimilarity;DiceSimilarity;DynamicTimeWarpingDistance;InnerProductSimilarity;JaccardSimilarity;KernelEuclideanDistance;ManhattanDistance;MaxProductSimilarity;OverlapSimilarity.

This results in more than 50 particular runs of the clustering algorithm. It seems that the parameter which primarily affects the quality of clustering is the number of clusters. The clustering quality (in terms of the Davies–Bouldin index) improves with an increasing number of clusters. We achieved the best results (lowest Davies–Bouldin index) for clustering into six clusters by using the correlation similarity measure.

The associations between the clusters and clinical features were assessed. The variables which presented a normal distribution were described as a mean ± standard deviation, and the non-normal variables were presented as medians and interquartile ranges. The categorical variables were shown as numbers and percentages (Table 2). The normality of the distribution was checked using the K–S, Lilliefors and Shapiro–Wilk tests. The statistical significance of differences between groups was assessed using analysis of variance, chi-square and ANOVA. The outcome associations were evaluated using the Kaplan–Meier curves and Cox proportional-hazards regressions (Figure 1). A *p*-value below 0.05 was considered statistically significant. Clustering and preprocessing were performed using RapidMiner 9.1 (RapidMiner GmbH, Dortmund, Germany) , and the statistical analysis was performed using STATISTICA 12 ((StatSoft Polska Sp. z o.o., Krakow, Poland)).

## 3. Results

### 3.1. Patients Characteristics

The study population consisted of 381 patients (all Caucasian), predominantly male 285 (75%), mean age 68 (60–79), with a median EF of 33% (25–45) and a median NTproBNP of 5580 (3169–10421) pg/mL (Table 2). The analyzed cohort presented as median: systolic blood pressure: 130 (110–150) mmHg, serum Na+: 139 (136–142) mmol/L and serum creatinine 1.23 (1–1.51) mg/dL. Table 3 shows the patient characteristics, including the key clinical features of each cluster. The principal clinical, biochemical and echocardiographic features of each cluster are presented in Figure 2.

### 3.2. Clustering

The population was divided into six cluster groups by analysis of 63 variables. Clusters have been enumerated from 0 to 5. The variables that were included in the analyses are presented in Table 1.

Cluster 0 (n = 86)

This was the largest cluster and included the highest percentage of patients with HF de novo, qualified as NYHA I, presenting with severe lower extremity edema on admission, and the highest urine K+, creatinine and urea levels. Moreover, this cluster had the highest ferritin levels and the lowest percentage of patients with a history of stroke.

Cluster 1 (n = 50)

Among the other clusters, this cluster was mostly represented by women with the highest prevalence of hypertension, diabetes, COPD and stroke history. On admission, this cluster presented with the highest systolic blood pressure, the highest percentage of patients with severe pulmonary congestion and the least severe signs of peripheral congestion. The NTproBNP, GFR, urine K+, urea and creatinine levels were the lowest in this cluster.

Cluster 2 (n = 70)

On average, this cluster was represented by the youngest patients, the highest percentage of active smokers, and qualified as NYHA IV and HF etiology was classified as other. Additionally, this cluster presented the lowest percentage of ischemic HF etiology, hypertension, diabetes and pulmonary congestion. On admission, they presented with the highest GFR, NTproBNP, AST, ALT and bilirubin serum levels and the lowest levels of troponin, CRP, and IL-6 serum levels.

Cluster 3 (n = 71)

This cluster consisted of the highest percentage of patients who qualified as HFrEF, the highest percentage of patients who decompensated in CHF, and the highest ratio of patients with valvular heart disease. On admission, this cluster was represented by the highest proportion of patients presenting with the most severe pulmonary congestion and lowest WBC, ferritin, TSAT, urine Na+, lactates and highest troponin, INR and albumin in the laboratory measurements.

Cluster 4 (n = 50)

This cluster was mostly represented by men, smokers with a CAD HF etiology and the lowest EF. On admission, they presented with the highest ratio of hepatomegaly, ascites, the highest JVP and the least frequent severe pulmonary congestion. They also had the highest creatinine and urea serum levels. For the arterial blood gases, this cluster presented with the lowest pCO_2_ and the highest pH.

Cluster 5 (n = 54)

The characteristics of these patients appeared to be the oldest population, with the highest percentage of women and the highest EF, lowest body mass, and no CAD history. Moreover, the highest level of CRP and Il-6 serum levels was in this group of patients.

### 3.3. Prognostic Significance of Clusters

The one-year mortality was 27% (104 events). The mean hospital stay was 8.6 ± 6.7 days.

The one-year mortality from cluster 0 to cluster 5 was: 26% vs 22% vs 17% vs 21% vs 40% vs 43%, *p* = 0.002, respectively (Table 4).

The risks for one-year compared with the rest of the population were calculated for each cluster. Clusters 5 and 4 had the highest one-year mortality risks, hazard ratio (95% confidence interval); cluster 5 had a HR (95% CI): 2.095 [1.327–3.306], *p* = 0.002; cluster 4: HR (95% CI): 1.738 [1.067–2.831], *p* = 0.026. Cluster 2 had the lowest one-year mortality risk, HR (95% CI): 0.537 [0.294–0.979], *p* = 0.043. There were no significant differences compared to the rest of the population for clusters 0, 1 and 3 (Table 5).

Figure 3 shows the Kaplan–Meier curves for the one-year mortality risks by clusters.

## 4. Discussion

A cluster analysis was applied to the cohort of 381 AHF patients. Both the clinical and biochemical variables were included and were either continuous or numerical. When writing this article, this was the most numerous analysis of such a type done in a European AHF population. Six clinically and pathophysiological relevant phenotypes were distinguished. The clusters varied in outcomes, including mortality and AHF re-hospitalization rates. Notably, the number of groups has not been prespecified, as in previous papers on the AHF population [1], but mathematically assessed. The quantity of the analyzed population allowed us to distinguish the highest number of virtually equally dense clusters [4,8,11,12,13], which provide the most thorough insight into an AHF’s population heterogeneity. Although during the collection of both registries guidelines for the treatment of heart failure have changed and a variety of new drugs have been implemented in therapy, such as angiotensin receptor neprilysin inhibitor, sodium-glucose co-transporter-2 inhibitors, a new class of beta-blockers and mineralocorticoid receptor antagonists, distinguished clusters seem to be resistant to that changes, because we have not included pre-admission treatment into cluster analysis. The decision above was dictated by practical reasons. According to the characteristics of the studied population and the numerous comorbidities with their special treatment, the quality of the analysis would not have been enhanced by including them. Therefore, the new drugs and guidelines are very unlikely to impact our cluster analysis, especially in terms of the cluster composition, which was based on the clinical and biochemical profiles at admission. The new guidelines would rather impact the patients’ prognosis. Noteworthy, these changes had very little, if any, impact on the outcomes of the population of patients with AHF. The one-year mortality of the studied population was 27%, which is not very distant from the current numbers (25–30%) [1]. Below, we present a detailed description of the clusters grouped according to their distinguishing clinical feature.

### 4.1. Clusters 1 and 4

Clusters 1 and 4 included patients with a high number of cardiovascular and non-cardiovascular comorbidities. In both these groups, coronary artery disease was the predominant etiology of heart failure.

Although these two clusters demonstrated similarities in terms of etiology, their prognosis and clinical outcome were significantly different. Cluster 1 had a relatively good prognosis, while cluster 4 had a poor clinical outcome. The one-year mortality was equal to 22% in cluster 1 and was almost twice as high in cluster 4 (40%), which can be explained by two factors.

The results of this study indicate that gender has a significant impact on the development of coronary artery disease and the progression of heart failure. It is well known that the male gender is itself a risk factor for cardiovascular events, and the prevalence of cardiovascular disease is higher in men than in women of a similar age [14].

In the case of the male population, the risk of cardiovascular disease increases linearly over time, and the atherosclerotic process develops continuously. On the other hand, due to the protective role of estrogen and its beneficial effects on the cardiovascular system, women of a fertile age may be protected from atherosclerosis [15,16,17,18]. This statement is consistent with our observations. Despite many risk factors, only 56% of patients from cluster 1(female-dominated) developed CAD, and only 40% had an MI. These values were significantly higher in the male-dominated population represented by cluster 4.

Additionally, as is commonly known, the incidence of stroke increases significantly in the postmenopausal period [19,20]. This also aligns with our observations, as the highest rate of stroke was reported for cluster 1.

It can, therefore, be inferred that gender plays a significant role in the development of cardiovascular diseases, and we assume that the differences in prognosis and clinical outcomes between these two groups could be partially explained by this fact. However, what determines the differences between these two clusters’ prognoses, for the most part, is their renal function.

Importantly, the phenotype of cluster 4 reflects the common problem of cardiorenal syndrome (the highest mean value of creatinine (1.36) and urea (64)) and right ventricular failure with the highest incidence of ascites, JVP and hepatomegaly, which constitute a sign of congestion. Cardiorenal syndrome and volume overload are well-documented predictors of poor outcomes [21] and are strongly associated with each other. Therapy for heart failure patients with cardiorenal syndrome remains a challenge. Its main goal should be reasonable decongestion, which can be achieved by natriuresis-guided diuretic therapy, ultrafiltration, or, in the refractory cases, experimental techniques.

### 4.2. Cluster 2

Patients included in cluster 2 were the youngest (mean age 58.8) and had the highest NTproBNP (7189), bilirubin (1.25), Ast (30), and Alt (34.5), and had the lowest ejection fraction (28%), serum creatinine concentration (1,1) incidence of diabetes (19%), pulmonary congestion (17%), COPD (5.7%), HT (39%), CAD (1.4%) and MI in the past (1.4%). These patients constituted the highest percentage of active smokers (30%) and alcohol consumers (44%). The underlying cause of HF was mostly valvular (21%) or other (73%). We assume that the presented phenotype, especially the elevated concentration of liver enzymes and frequent tobacco and alcohol use, suggests a significant role in toxic myocardial damage. Importantly, cluster 2 was associated with the most positive prognosis. It can be explained by the youngest age, low morbidity, and high potential compensatory reserves. Therefore, these patients represent great therapeutic potential, and clinicians should focus on education in the context of eliminating the harmful impact of xenobiotics.

### 4.3. Cluster 3

The distinguishing features of cluster 3 (n = 71) were the incidence of chronic heart failure (93%) and iron deficiency

The prevalence of iron deficiency (defined as a serum ferritin < 100 ng/mL or TSAT < 20) [22] is common in this population. In comparison to the other groups, cluster 3 represented the lowest mean value of ferritin (92) and TSAT (14.8%).

Iron deficiency is a frequent comorbidity in heart failure, present in approximately 30–50% of patients and is associated with worse long-term outcomes [23,24,25].

The detrimental effect of imbalanced iron homeostasis on HF progression has been widely studied; however, it remains unclear what the exact mechanism is by which an iron deficit worsens HF. It appears that there is a wide range of factors involved in this process.

First of all, iron deficiency alters mitochondrial function and impairs the already disturbed energetics of the heart with a reduced ejection fraction [26].

Secondly, in the condition of iron deficiency anemia, depleted oxygen delivery to the metabolizing tissues induces a variety of hemodynamic, renal, and neurohormonal alterations [27]. Volume expansion (caused by sympathetic and RAA activation), as well as vasodilatation, leads to an increase in cardiac output. All these mechanisms result in an increased myocardial workload and further hypertrophy/remodeling of LV, which contributes to worsening HF [28].

We assume that the iron deficiency could explain the mean prognoses of patients in this cluster and constitute a relatively easy-achievable therapeutic goal to improve these patients’ outcomes.

### 4.4. Cluster 1 and Cluster 5

Both clusters 1 and 5 include mostly elderly (mean age 76.1 in both clusters) women (46% and 44%, respectively), with the highest ejection fraction (47.5% and 50%) and a high incidence of hypertension (94% and 87%). The presented phenotype corresponds to the well-established HFpEF patient characteristics [29]. The clusters present the most frequent incidence of massive pulmonary congestion (congestion auscultated over two-thirds of the lungs in the 22% and 13%), which is reflected by the highest proportion of the NYHA IV (54% and 57%). The highest mean values of the pCO_2_ (37.2 mmHg and 36.2 mmHg) reflect the most massive pulmonary oedema or the relatively high incidence of lung comorbidities, especially COPD in the HFpEF population [30]. Despite the apparently similar phenotypes, the clusters significantly differed in outcomes (Figure 3). Cluster 1 presented a relatively good prognosis, and conversely, cluster 5 was associated with an ominous outcome. The features that especially differ the clusters are the types of HF (chronic/de novo) and the concentration of the inflammatory markers. Cluster 1 consisted mostly of the patients who presented with their first episode of HF (56%), and cluster 5 were the patients suffering from chronic HF (70%). The duration of HF is a well-established prognostic factor. Moreover, cluster 1 presented the highest natriuresis, probably due to the effect of the first presentation of HF and frequent loop-diuretics naiveness and, as a sign of adequate diuretic response, predicted a favorable outcome [31]. Subsequently, cluster 5 presented with the highest mean concentration of inflammatory biomarkers—CRP and Il-6, which, with the lowest mean body weight (74.9), suggests frailty syndrome and explains the poor prognosis [32,33].

### 4.5. Novelty and Clinical Implications

We presume that our paper has two significant advantages over the currently published clustering-based analysis of acute heart failure populations. Noteworthy, it is, by now, the most numerous clustering analysis for a European AHF population. Moreover, we have not prespecified the number of the clusters in advance, in order to allow the algorithm to distinguish the optimal, natural number of different subgroups autonomously.

The clustering technology is currently far from being an ideal solution for heart failure phenotyping. Nevertheless, we strongly believe that this technique presents great potential as a tool which can capture the relationships which are too complex to be noticed by a classical statistical analysis but can be visible to the experienced clinician. We believe it will eventually immediately segregate admitted HF patients into previously described groups (clusters). Such a segregation will highlight the therapeutic aspects that clinicians should focus on (e.g., cardiorenal syndrome, iron deficiency, etc.) and initially estimate a prognosis. Further, the patients who would be placed into the group with a worse prognosis could be provided with more careful/insightful treatment from the very beginning of the therapeutic process. For example, clusters 2 and 3 revealed the recognized relationships between HF and, consequently, chronic intoxication and iron deficiency. The precise outpatient care for the cluster 3 patients, with a regular iron level assessment and intravenous supplementation if needed, could reduce the likelihood of HF deterioration [25] Further, proper education and providing cluster 3 patients with specialist psychiatric care regarding their addiction and substance abuse could slow the progress of HF [34]. Noteworthy, the clustering, in that case, does not reveal relationships that are astounding for the experienced cardiologist. The potential value of such algorithms and provided classifications is its ability to immediately categorize the patients into one of the pheno-groups and underline cluster-specific treatment targets which can be accidentally omitted due to, e.g., the doctors’ overwork, overfilling the hospitals or the lack of experience of medical professionals.

## 5. Limitations

Our study is not free from limitations. First, the study included retrospective data. Therefore, the availability of potentially important clinical parameters was restricted. Variables, such as the echocardiographic parameters, invasive hemodynamic measurements or novel experimental markers, were not collected. New cluster-based trials with a broader biochemical and clinical composition would deliver exciting data. Moreover, the gathered data contained missing values. Notwithstanding restricting the data inclusion to 10% of the missing values, some bias could occur. Second, our analysis was based on single-center data from Poland, which included a relatively small sample size and lacked an external validation cohort. Consequently, the evaluated patients were treated following outdated guidelines. The current clinical presentation of AHF patients and their outcomes can differ from the presented results.

The machine-learning techniques can be associated with the overfitting problem, in which the model performs well on the seen population and poor on the new one. In other words, the model is not generalizable. However, in unsupervised learning, to which clustering belongs, there is no such information about a “true” or “correct” assignment of examples to clusters. The clustering only works with the given data, and the possible generalization of the clusters is rather a question of their interpretation by the domain (medical) experts rather than a question of evaluation on another dataset. Thus, overfitting, in the standard sense, is not an issue for clustering.

What is important in clustering is having a “reasonable” number of clusters. The small number of clusters will produce over-general results—the worst case is just one cluster for everything, a large number of clusters will produce over-specific results—the worst case is that each example creates its own cluster. The problem of over-specific results can be, in some sense, considered similar to the problem of overfitting—we tried to avoid it by automatically tuning the range of clusters. We designed the range of the number of possible clusters from three to six to avoid excessive data fragmentation. Such an approach was consistent with prior studies, which usually consisted of three to four groups [4,8,12,13]. We decided to increase the potential number of the clusters due to the bigger included population. Further analyses to highlight more nuanced phenotypes are warranted.

## 6. Conclusions

We successfully extracted six novel phenotypes of acute heart failure patients, providing a fresh insight into their heterogeneity. The proposed clusters were consistent with the latest understanding of pathophysiology (e.g., de novo HF, HT HFpEF, toxic HF, iron reduced left ventricle HF, cardiorenal, inflammatory HFpEF) and previous clustering-based papers, providing a more distinctive classification of the population. Presented results can be valuable for future AHF trial constructions and more customized treatments.

## Figures and Tables

**Figure 1 biomedicines-10-01514-f001:**
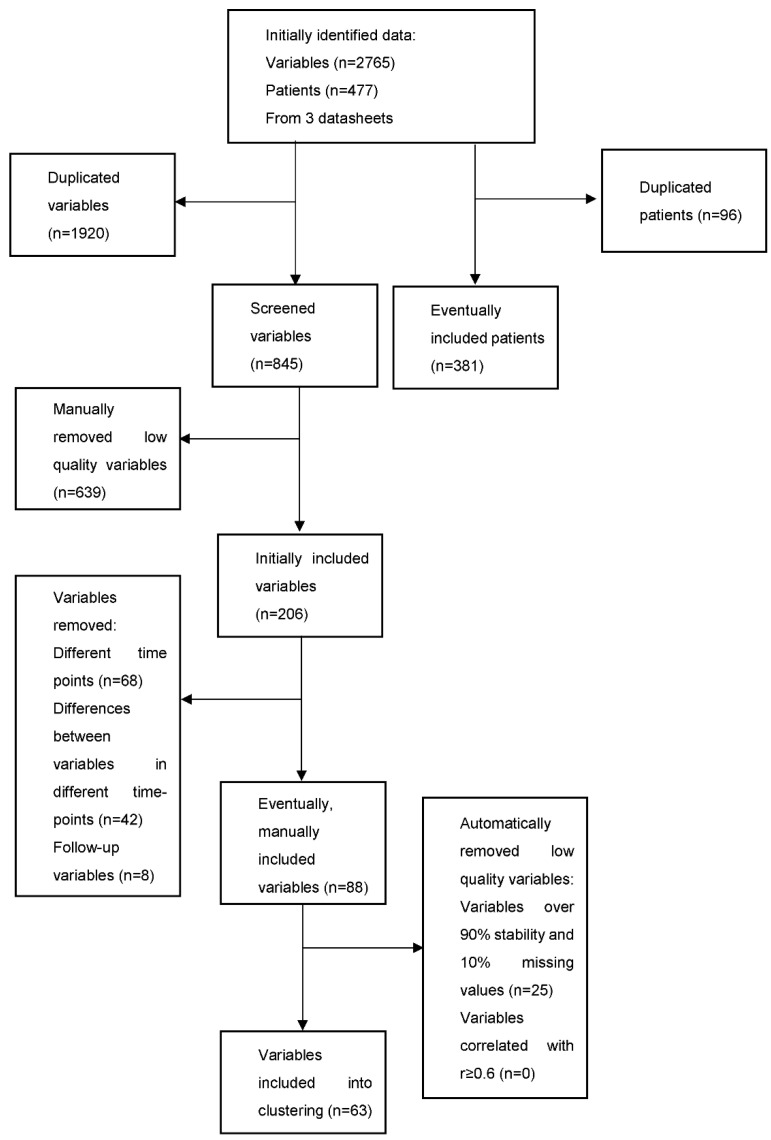
Flowchart of the analyzed variables and patients. The analysis was conducted based on the previously prepared data, therefore, some of the information was duplicated or inadequate for the machine-learning analysis.

**Figure 2 biomedicines-10-01514-f002:**
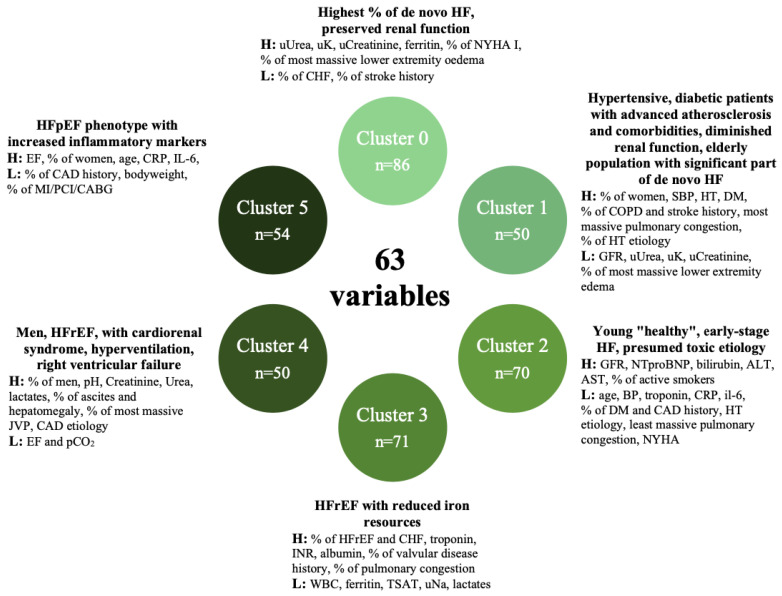
Principal clinical, laboratory and echocardiographic features for each cluster. ALT—Alanine Aminotransferase, AST—Aspartate Aminotransferase, BP—blood pressure, CABG—coronary artery bypass grafting, CAD—coronary artery disease, CHF—chronic heart failure, COPD—chronic obstructive pulmonary disease, CRP—C-reactive protein, DM—diabetes mellitus, EF—ejection fraction, GFR—glomerular filtration ratio, HF—heart failure, HFrEF—heart failure with reduced ejection fraction, HT—hypertension, IL-6—interleukin 6, JVP—jugular venous pulsation, MI—myocardial infarction, NTproBNP—N-terminal-pro B-type natriuretic peptide, NYHA—New York Heart Association class, PCI—percutaneous cardiac intervention, SBP—systolic blood pressure, TSAT—Transferrin saturated with iron, u—urine concentration, WBC—white blood cell count.

**Figure 3 biomedicines-10-01514-f003:**
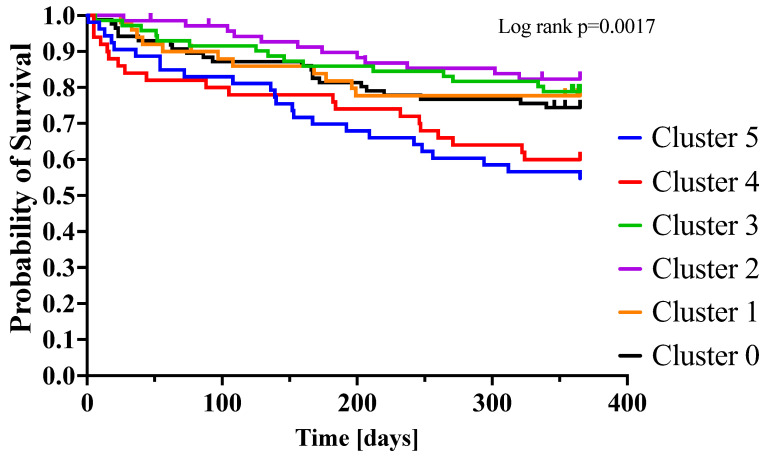
Kaplan–Meier curves for one-year mortality by clusters.

**Table 1 biomedicines-10-01514-t001:** Variables initially included in the analysis. All parameters were assessed at admission. Bolded variables are variables which were included in the cluster analysis after the automatic preprocessing.

Demographics	**Age, Sex**
HF characteristics	**De novo or chronic HF, Etiology**
Comorbidities	**Coronary artery disease (0 or 1), myocardial infarction (0 or 1), PCI/CABG (0 or 1)**, **hypertension (0 or 1), valvular heart disease (0 or 1), diabetes (0 or 1), diabetes treated with: insulin = 1** **Oral drugs = 2, diet = 3, stroke (0 or 1),COPD (0 or 1)**
Clinical status	**Dyspnoea at rest (0 or 1),** Dyspnoea at rest lasts since (number) days, **NYHA at admission, swelling of the lower limbs (lack = 0, 1 + (10–15 s) = 1, 2 + (15–30 s) = 2, 3 + (>30 s) = 3),** Decrease in exercise tolerance (0 or 1), decrease in exercise tolerance (for how many days), body weight, **systolic pressure, diastolic pressure, heart rate, jugular veins pressure (<6 cm = 1, 6–10 cm = 2, >10 cm = 3, not to be assessed = 4)**, **pulmonary congestion (no—0; up to 1/3 of lungs—1; up to 2/3—2; >2/3—3)**, **pulmonary congestion (0 or 1)**, **ascites (0 or 1), hepatomegaly (0 or 1), implantable device, none = 0, 1-PM, 2-ICD, 3-CRT2**
Lifestyle factors	**Smoking status (0 = never, 1 = now, 2 = in the past)**. If smoking in the past, how many cigarettes did the patient smoke? **Alcohol (0 or 1)**, How many cigarettes do the patients smoke daily, How many years did the patient smoke/does the patient smoke cigarettes?
Laboratory parameters	**HGB, HCT, RBC, MCV, MCH, MCHC, RDW, WBC, LYMPH, MONO, NEUTR, PLT,** serum PH, pCO_2_, pO_2_, ctO_2_, BO_2_, HCO_3_, HCO_3_std, ctCO_2_, BE, sO_2_, FO_2_Hb, FHHb, ctHb, Lac, mOsm, **Na in serum, K in serum, Creatinine in serum, Urea in serum, Glucose in serum, Ast, Alt**, **CRP,** GGTP, **NTproBNP, Total_bilirubin, INR, Albumin in serum, Troponin in serum, Urine Na, Urine K, Urine Urea, Urine Creatinine, Fe, TIBC, Tsat, sTfR,** Ferritin, **IL-6, eGFR**
Echocardiography	**Reduced ejection fraction (0 or 1);** ejection fraction

Abbreviations: HGB—hemoglobin, HCT—hematocrit, RBC—red blood count, MCV—mean corpuscular volume, MCH—mean corpuscular hemoglobin, MCHC—mean corpuscular hemoglobin concentration, RDW—red cell distribution width, WBC—white blood count, LYMPH—lymphocytes percentage, MONO—monocytes, NEUTR—neutrophiles, PLT—platelets count, pCO_2_—partial pressure of CO_2_, pO_2_—partial pressure of O_2_, ctO_2_—concentration of O_2_, BO_2_ -, HCO_3_- bicarbonate, HCO_3_std—bicarbonate standardized, ctCO_2_—CO_2_ concentration, BE—base excess, sO_2_—O_2_ saturation, FO2Hb—fraction of oxygenated haemoglobin, FHHb—fraction of deoxyhemoglobin in total hemoglobin, ctHb—total hemoglobin, Lac—lactates, mOsm -milliosmoles, Ast—aspartate aminotransferase, Alt—alanine transaminase, CRP—C-reactive protein, GGTP—gamma-glutamyl transpeptidase, NTproBNP—N-terminal prohormone of brain natriuretic peptide, INR—international normalized ratio, Fe—total iron amount in blood, TIBC—total iron-binding capacity, Tsat—transferrin saturation, sTfR—Soluble Transferrin Receptor, IL-6—interleukin 6th, eGFR—estimated glomerular filtration rate.

**Table 2 biomedicines-10-01514-t002:** Characteristics stratified by clusters and in the whole group. The highest values of the variables are marked red, lowest ones are green.

Parameter	Cluster_0	Cluster_1	Cluster_2	Cluster_3	Cluster_4	Cluster_5	Global	*p*
Demographics								
n	86	50	70	71	50	54	381	-
Sex, male (n)	78 (90.698%)	23 (46%)	58 (82.857%)	53 (74.648%)	49 (98%)	24 (44.444%)	285 (74.803%)	<0.001
Age (years)	67.293 [59–79]	76.1 [68–81]	58.821 [51.279–67.003]	72 [63–80]	66 [60.29–74.521]	76.111 [64–82.992]	68 [60–79]	<0.001
aHF charcteristics								
Ejection fraction	34 [28–43]	47.5 [39–55]	28 [20–40]	30 [25–35]	28 [20–35]	50 [30–60]	33 [25–45]	<0.001
Chronic HF (n)	32 (37.209%)	22 (44%)	34 (48.571%)	69 (97.183%)	47 (94%)	38 (70.37%)	242 (63.517%)	<0.001
Reduced EF (n)	67 (77.907%)	16 (32%)	58 (82.857%)	66 (92.958%)	45 (90%)	17 (31.481%)	269 (70.604%)	<0.001
Etiology								<0.001
Coronary artery disease (n)	41 (47.674%)	28 (56%)	3 (4.286%)	61 (85.915%)	43 (86%)	20 (37.037%)	178 (46.719%)	
Valvular (n)	5 (5.814%)	2 (4%)	15 (21.429%)	3 (4.225%)	1 (2%)	2 (3.704%)	46 (12.073%)	
Hypertension (n)	1 (1.163%)	5 (10%)	1 (1.429%)	1 (1.408%)	1 (2%)	4 (7.407%)	13 (3.412%)	
Other (n)	39 (45.349%)	15 (30%)	51 (72.857%)	6 (8.451%)	5 (10%)	28 (51.852%)	144 (37.795%)	
Comorbidites								
Coronary artery disease (n)	56 (65.116%)	38 (76%)	1 (1.429%)	69 (97.183%)	49 (98%)	5 (9.259%)	218 (57.218%)	<0.001
Myocardial infarction in the past (n)	17 (19.767%)	20 (40%)	1 (1.429%)	33 (46.479%)	44 (88%)	3 (5.556%)	118 (30.971%)	<0.001
PCI/CABG in the past (n)	9 (10.465%)	27 (54%)	0 (0%)	50 (70.423%)	37 (74%)	0 (0%)	123 (32.283%)	<0.001
Hypertension (n)	72 (83.721%)	47 (94%)	27 (38.571%)	56 (78.873%)	38 (76%)	47 (87.037%)	286 (75.066%)	<0.001
Valvular disease (n)	52 (60.465%)	16 (32%)	43 (61.429%)	57 (80.282%)	38 (76%)	38 (70.37%)	244 (64.042%)	<0.001
Diabetes mellitus (n)	30 (34.884%)	46 (92%)	13 (18.571%)	22 (30.986%)	27 (54%)	14 (25.926%)	152 (39.895%)	<0.001
Diabetes treatment (n)								
Insulin	5 (5.814%)	20 (40%)	1 (1.429%)	7 (9.859%)	9 (18%)	1 (1.852%)	43 (11.286%)	
Oral drugs	11 (12.791%)	17 (34%)	7 (10%)	10 (14.085%)	13 (26%)	11 (20.37%)	69 (18.11%)	
Diet	5 (5.814%)	4 (8%)	0 (0%)	1 (1.408%)	4 (8%)	0 (0%)	14 (3.675%)	
Stroke (n)	7 (8.14%)	11 (22%)	8 (11.429%)	12 (16.901%)	9 (18%)	6 (11.111%)	53 (13.911%)	<0.001
COPD (n)	8 (9.302%)	11 (22%)	4 (5.714%)	12 (16.901%)	8 (16%)	7 (12.963%)	50 (13.123%)	<0.001
Clinical status								
Dyspnoea at rest (n)	76 (88.372%)	42 (84%)	40 (57.143%)	56 (78.873%)	43 (86%)	50 (92.593%)	307 (80.577%)	<0.001
Dyspnoea at rest lasts since (n) days	3 [1–8]	3 [1–7]	3.5 [1–8.5]	3 [2–8.5]	3 [2–7]	3 [2–6]	3 [1–7]	0.8
Decrease in exercise tolerance (n) days	14 [7–21]	7 [6.5–29]	14 [7–29]	14 [7–28]	10 [7–21]	14 [6.5–30]	14 [7–28]	0.6
NYHA (n)								0.243
I	4 (4.651%)	1 (2%)	3 (4.286%)	2 (2.817%)	2 (4%)	1 (1.852%)	13 (3.412%)	
II	11 (12.791%)	8 (16%)	13 (18.571%)	7 (9.859%)	13 (26%)	10 (18.519%)	62 (16.273%)	
III	12 (13.953%)	8 (16%)	23 (32.857%)	26 (36.62%)	9 (18%)	9 (16.667%)	87 (22.835%)	
IV	46 (53.488%)	27 (54%)	23 (32.857%)	36 (50.704%)	26 (52%)	31 (57.407%)	189 (49.606%)	
Swelling of lower limbs (n)								0.006
Swelling of lower limbs 0	18 (20.93%)	16 (32%)	26 (37.143%)	19 (26.761%)	16 (32%)	7 (12.963%)	102 (26.772%)	
Swelling of lower limbs 1	15 (17.442%)	15 (30%)	16 (22.857%)	18 (25.352%)	10 (20%)	16 (29.63%)	90 (23.622%)	
Swelling of lower limbs 2	27 (31.395%)	13 (26%)	17 (24.286%)	23 (32.394%)	11 (22%)	16 (29.63%)	107 (28.084%)	
Swelling of lower limbs 3	26 (30.233%)	6 (12%)	10 (14.286%)	11 (15.493%)	13 (26%)	15 (27.778%)	81 (21.26%)	
Deterioration of Effort Tolerance (n)	79 (91.86%)	47 (94%)	63 (90%)	67 (94.366%)	49 (98%)	53 (98.148%)	358 (93.963%)	0.407
JVP (n)								<0.001
JVP 1	57 (66.279%)	32 (64%)	42 (60%)	53 (74.648%)	17 (34%)	31 (57.407%)	232 (60.892%)	
JVP 2	24 (27.907%)	17 (34%)	23 (32.857%)	18 (25.352%)	25 (50%)	21 (38.889%)	128 (33.596%)	
JVP 3	5 (5.814%)	0 (0%)	5 (7.143%)	0 (0%)	8 (16%)	2 (3.704%)	20 (5.249%)	
Pulmonary edema (n)								<0.001
no	11 (12.791%)	1 (2%)	12 (17.143%)	2 (2.817%)	7 (14%)	6 (11.111%)	39 (10.236%)	
up to 1/3 of lungs	49 (56.977%)	23 (46%)	45 (64.286%)	50 (70.423%)	31 (62%)	25 (46.296%)	223 (58.53%)	
up to 2/3	20 (23.256%)	14 (28%)	9 (12.857%)	13 (18.31%)	11 (22%)	16 (29.63%)	83 (21.785%)	
>2/3	6 (6.977%)	11 (22%)	4 (5.714%)	6 (8.451%)	1 (2%)	7 (12.963%)	35 (9.186%)	
Pulmonary congestion (n)	75 (87.209%)	48 (96%)	58 (82.857%)	69 (97.183%)	43 (86%)	48 (88.889%)	341 (89.501%)	0.048
Ascites (n)	15 (17.442%)	3 (6%)	9 (12.857%)	2 (2.817%)	13 (26%)	8 (14.815%)	50 (13.123%)	0.003
Hepatomegaly (n)	29 (33.721%)	8 (16%)	11 (15.714%)	1 (1.408%)	27 (54%)	6 (11.111%)	82 (21.522%)	<0.001
Implantable device (n)								<0.001
PM	2 (2.326%)	8 (16%)	2 (2.857%)	8 (11.268%)	2 (4%)	6 (11.111%)	28 (7.349%)	
ICD	3 (3.488%)	1 (2%)	8 (11.429%)	31 (43.662%)	9 (18%)	3 (5.556%)	55 (14.436%)	
CRT	2 (2.326%)	1 (2%)	3 (4.286%)	3 (4.225%)	15 (30%)	2 (3.704%)	26 (6.824%)	
Systolic pressure (mmHg)	140 [120–158]	160 [135–180]	120 [105–131]	126.5 [110–137]	120 [102–145]	120 [107–142]	130 [110–150]	<0.001
Diastolic pressure (mmHg)	80 [70–95.5]	80 [70–95]	77.5 [70–87]	80 [70–85]	70 [62–80]	70 [65–80]	79 [70–90]	<0.001
Heart rate (bpm)	90 [75–110]	80 [70–100]	90.5 [80–105]	80 [70–100]	78 [70–90]	88 [72–110]	82.5 [70–100]	<0.001
Body weight (kg)	85.3 [77–98]	79 [69–90.95]	77.6 [68.5–88.3]	77.4 [70.4–91]	80.5 [71–94]	74.9 [65–82]	79.6 [70–91.5]	<0.001
Lifestyle factors								
Smoking status (n)								<0.001
Never	41 (47.674%)	32 (64%)	35 (50%)	49 (69.014%)	8 (16%)	36 (66.667%)	201 (52.756%)	
Active	23 (26.744%)	3 (6%)	21 (30%)	7 (9.859%)	4 (8%)	3 (5.556%)	61 (16.01%)	
In the past	22 (25.581%)	15 (30%)	14 (20%)	15 (21.127%)	38 (76%)	15 (27.778%)	119 (31.234%)	
How many cigarettes do patients smoke daily (n)	0.08 [0–15]	1 [0–8]	0 [0–15]	0 [0–9]	15 [4–20]	3 [0–12]	2 [0–15]	0.047
How many years did the patient smoke/does the patient smoke cigarettes (n)	22.5 [0–30]	20 [0–30]	11.5 [0–30]	0 [0–30]	20 [5–30]	0 [0–30]	20 [0–30]	0.36
Active alcohol use (n)	20 (23.256%)	8 (16%)	31 (44.286%)	16 (22.535%)	19 (38%)	12 (22.222%)	106 (27.822%)	0.002
Laboratory parameters								
HGB (g/dL)	13.727 ± 1.881	11.972 ± 1.81	13.975 ± 1.651	13.213 ± 1.817	13.194 ± 2.114	12.391 ± 1.801	13.184 ± 1.953	<0.001
HCT (%)	41.232 ± 5.21	36.686 ± 5.191	41.684 ± 4.665	39.907 ± 5.163	40.066 ± 6.319	37.343 ± 4.854	39.759 ± 5.49	<0.001
RBC (× 10^12^/L)	4.544 ± 0.662	4.18 ± 0.55	4.595 ± 0.495	4.499 ± 0.65	4.516 ± 0.716	4.226 ± 0.628	4.448 ± 0.636	<0.001
MCH (pg)	30.333 ± 2.325	28.692 ± 2.728	30.457 ± 2.269	29.49 ± 2.261	29.255 ± 2.565	29.479 ± 2.986	29.718 ± 2.552	<0.001
MCV fL	91.188 ± 6.241	87.846 ± 6.236	90.854 ± 5.707	89.057 ± 6.144	89.034 ± 6.797	88.834 ± 6.451	89.668 ± 6.31	0.02
WBC (× 10^9^/L)	8.6 [6.8–10.68]	9.35 [6.7–12.3]	8.25 [6.3–9.85]	7.8 [6.4–9.52]	8.44 [7.1–10.4]	8.3 [6.1–9.9]	8.3 [6.6–10.35]	0.01
PLT (× 10^9^/L)	214 [152–252.5]	211 [163–298]	197.5 [164.5–233]	192 [149–234]	195 [159–250]	203 [144–242]	198 [155–245]	0.04
pH	7.44 [7.415–7.47]	7.4 [7.35–7.46]	7.45 [7.42–7.48]	7.45 [7.43–7.47]	7.45 [7.415–7.485]	7.45 [7.385–7.48]	7.44 [7.41–7.47]	<0.001
pCO2 (mmHg)	34.4 [31.55–38.7]	37.3 [32.7–42.9]	34.55 [30.9–36.55]	34.55 [32.2–37.5]	33.6 [31.6–38.25]	36.2 [33.05–39.45]	35.1 [31.8–38.9]	<0.001
HCO3std (mmol/L)	24.016 ± 3.193	22.989 ± 3.657	24.592 ± 2.474	24.676 ± 2.684	24.602 ± 3.376	25.321 ± 3.688	24.367 ± 3.203	0.01
pO2 (mmHg)	64.4 [57.15–73.15]	66.3 [61.2–78.7]	70.2 [62.3–75.5]	65.6 [58.2–74.3]	67.3 [60.05–74.7]	65.15 [57.65–71.8]	66.1 [59–74.6]	0.8
sO2 (%)	92.1 [89.15–95.05]	93.45 [90.6–94.9]	94.45 [91.45–95.95]	92.8 [89.9–94.9]	93.1 [90.4–96]	93.05 [90.2–95.4]	93.1 [90.1–95.4]	0.9
mOsm (Osm/L)	282.5 [274–286]	286.5 [279–291]	283 [274–287]	281 [274–286]	277.5 [272–286]	279.5 [270–287]	282 [274–287]	0.01
K (mmol/L)	4.187 ± 0.577	4.481 ± 0.788	4.197 ± 0.484	4.185 ± 0.521	4.197 ± 0.622	4.063 ± 0.694	4.21 ± 0.614	0.02
Na (mmol/L)	140 [137–142]	140 [137–142]	139 [135.5–141.5]	139 [137–142]	138 [135–140]	138.5 [135–141]	139 [136–142]	0.145
Glucose (mg/dL)	124 [100–162]	144 [121–212]	110 [99.5–131]	113 [101–139]	126.5 [107–150]	117 [105–143]	121 [103–151.5]	<0.001
INR	1.26 [1.08–1.48]	1.31 [1.09–1.99]	1.31 [1.14–1.77]	1.54 [1.18–2.24]	1.42 [1.17–2.08]	1.46 [1.2–2.21]	1.35 [1.12–1.97]	0.06
Total bilirubin (mg/dL)	0.96 [0.72–1.46]	0.785 [0.505–1.275]	1.25 [0.765–1.755]	1.145 [0.775–1.945]	1.225 [0.855–1.705]	1.03 [0.79–1.9]	1.07 [0.73–1.7]	0.09
Albumin (g/dL)	3.675 ± 0.402	3.775 ± 0.342	3.755 ± 0.406	3.831 ± 0.328	3.766 ± 0.386	3.648 ± 0.466	3.739 ± 0.394	0.1
Ast (IU/L)	29 [21.5–44.5]	26 [17–37]	30 [22–40]	26 [20–37]	26.5 [18–34.5]	27 [20.5–38.5]	27 [20–40]	0.5
Alt (IU/L)	28 [21.5–58]	28 [17–41]	34.5 [21.5–55]	30.5 [21–53]	27.5 [16.5–40.5]	24.5 [15.5–32]	29 [19–48]	0.7
GGTP (IU/L)	70 [40–127]	54.5 [39.5–102.5]	82 [48–166]	72 [48–133]	104 [45–183]	60.5 [28–113.5]	71 [41–128]	0.8
TIBC (μg/dL)	331.45 ± 63.813	336.5 ± 84.925	362.968 ± 66.412	364.09 ± 68.448	366.302 ± 60.677	338.765 ± 72.717	349.457 ± 70.214	0.007
Fe (μg/dL)	48 [36–66.5]	47.5 [31.5–65.5]	60 [47–84]	55 [43–79]	62 [43–83]	50 [37–61]	54 [40–71]	0.009
Ferritin (ng/mL)	162.5 [85.325–252]	147.5 [57–249]	124 [52–224]	92 [54–156]	94.985 [53.68–146]	119.6 [67.36–200]	109.3 [61–224]	0.02
Tsat (%)	15.25 [10.113–20.1]	15.05 [9.263–19.057]	16.958 [13.2–25.455]	14.8 [11.4–21.4]	17 [12.429–23.4]	15.9 [12.4–18.3]	15.654 [11.609–21.05]	0.46
sTfR (mg/L)	1.72 [1.42–2.72]	2.02 [1.445–2.635]	1.73 [1.41–2.08]	1.97 [1.69–2.51]	1.905 [1.59–2.46]	1.79 [1.3–2.73]	1.87 [1.46–2.51]	0.66
NTproBNP (pg/mL)	5218 [2674–12496]	4191 [2025–6048]	7189 [5023–12849]	5437 [3612–10572]	5712.5 [3452.5–11170.5]	5337 [2398–8775]	5580 [3169–10421]	0.03
Troponin (ng/mL)	0.042 [0.022–0.12]	0.049 [0.025–0.156]	0.032 [0.017–0.094]	0.058 [0.03–0.156]	0.05 [0.029–0.13]	0.05 [0.02–0.14]	0.05 [0.022–0.127]	0.03
CRP (mg/L)	8.6 [4.4–19.3]	6.8 [3.05–27.25]	6.15 [3.2–14.05]	7.425 [3.8–14.5]	6.95 [3.25–16.05]	8.18 [3.86–19.4]	7.395 [3.5–18]	0.18
IL6 (pg/mL)	12.108 [4.428–26.822]	10.999 [0.633–27.125]	7.979 [0.5–19.923]	8.315 [0.5–14.6]	8 [4.851–16.927]	13.82 [3.785–38.5]	9.989 [2.528–22.89]	0.29
Lactates (mmol/L)	2 [1.4–2.4]	1.95 [1.5–2.7]	2 [1.6–2.7]	1.8 [1.5–2.4]	2.1 [1.45–2.7]	2 [1.5–2.75]	2 [1.5–2.6]	0.64
Urea (mmol/L)	47 [37–73]	55 [39–78]	49.5 [38–68]	53.5 [43–74]	64 [44–86]	44 [35–65]	51 [38–73]	0.3
Creatinine (mg/dL)	1.16 [1.03–1.5]	1.32 [0.93–1.7]	1.1 [0.935–1.295]	1.23 [1.03–1.49]	1.355 [1.09–1.8]	1.2 [0.95–1.44]	1.225 [1–1.505]	0.003
eGFR (mL/min/1.73m^2^)	84.463 ± 26.383	68.036 ± 29.564	94.693 ± 31.385	76.697 ± 22.711	77.859 ± 34.792	79.116 ± 43.668	81.074 ± 32.041	<0.001
Urine Urea (mmol/L)	1131 [555.5–1585]	512 [369–905]	886 [484–1674]	730 [442–1330]	887 [487–1509]	514 [339.5–981]	780 [442–1403]	<0.001
Urine Creatinine (mg/dL)	80.55 [41.75–147.6]	33.5 [21.7–79.2]	73.2 [34.7–129.1]	61.5 [28.9–105]	52.9 [38.9–136.8]	42 [23.55–80.65]	59.1 [30.1–110]	<0.001
Urine K (mmol/L)	35.765 [20–49.04]	22.75 [15–32]	28.73 [20–41]	27 [17.14–37]	31.5 [27–50.44]	29.5 [17–41.5]	29.77 [19–42.59]	<0.001
Urine Na(mmol/L)	87.286 ± 39.226	95.432 ± 32.757	90.87 ± 42.771	87.594 ± 37.329	84.533 ± 34.78	96.269 ± 36.412	89.959 ± 37.886	0.55

**Table 3 biomedicines-10-01514-t003:** Key clinical features of each cluster.

Cluster	Key Clinical Feature
Cluster 0	Lowest % of chronic HF, most massive lower limbs oedema, highest urine urea, k, creatinine, highest ferritin, highest % of NYHA I, lowest % stroke history, better prognosis—**highest % of de novo HF, with preserved renal function.**
Cluster 1	Higher % of women than in the rest of the population, highest systolic pressure, highest hypertension, diabetes, chronic obstructive pulmonary disease and stroke history (lowest GFR, lowest urine creatinine, urea and K, lowest NTproBNP), most massive pulmonary congestion and least massive peripheral oedema, highest hypertension etiology, better prognosis—**hypertensive, diabetic patients with advanced atherosclerosis and comorbidities, diminished renal function, elderly population with a significant part of de novo HF.**
Cluster 2	Youngest patients, low NYHA and ejection fraction, lowest blood pressure, troponin, CRP and IL-6, lowest % diabetes history, lowest % of CAD history and etiology, lowest hypertension etiology, highest “other” etiology, highest GFR, NTproBNP, bilirubin, Alt, Ast, highest % of active smokers, least massive pulmonary congestion, better prognosis—**young “healthy”, early-stage HF, presumed toxic etiology.**
Cluster 3	Lowest WBC, ferritin, urine Na, Tsat, lactates, highest troponin, INR, albumin, highest % of HFrEF and chronic HF, highest % of valvular disease history, highest % of pulmonary congestion (97%), mean prognosis—**HFrEF with reduced iron resources.**
Cluster 4	Predominantly man, highest pH, creatinine, urea, lactates, lowest ejection fraction and pCO_2_, highest % of ascites and hepatomegaly, most massive JVP, highest CAD etiology, worse prognosis—**men, HFrEF, with cardiorenal syndrome, hyperventilation, right ventricular failure.**
Cluster 5	Highest EF, no CAD history (0%), oldest population, highest % of women, highest CRP, IL6, lowest body weight, low % of MI/PCI/CABG, worst prognosis—**HFpEF phenotype with increased inflammatory markers.**

**Table 4 biomedicines-10-01514-t004:** Outcomes by Clusters.

	Cluster 5	Cluster 4	Cluster 3	Cluster 2	Cluster 1	Cluster 0	*p*
One-year mortality	45.3%	40%	21.1%	17.1%	22%	25.6%	0.002
One-year mortality or HF rehospitalization	68.1%	77.3%	55.7%	63.2%	55.3%	53.5%	0.112
In-hospital deterioration	8.5%	16.3%	8.2%	3.1%	15.2%	7.8%	0.1
Duration of hosp. [days]	9.3 ± 5.7	9.4 ± 6.8	6.7 ± 3.4	8.2 ± 7.5	9.7 ± 8.5	9.0 ± 7.3	0.1

**Table 5 biomedicines-10-01514-t005:** Hazard ratios for one-year mortality; each cluster was compared with the rest of the population.

	One-Year Mortality Risk		
	X^2^	Hazard Ratio (95% Confidence Interval)	*p*
Cluster 0	0.194	0.900 [0.562–1.441]	0.662
Cluster 1	0.679	0.776 [0.415–1.449]	0.425
Cluster 2	4.807	0.537 [0.294–0.979]	0.043
Cluster 3	1.964	0.688 [0.397–1.188]	0.179
Cluster 4	4.393	1.738 [1.067–2.831]	0.026
Cluster 5	8.753	2.095 [1.327–3.306]	0.002

## Data Availability

The data presented in this study are available within the article. Further data are available on request from the corresponding author.

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
