# Peer review of "Novel Phenotyping for Acute Heart Failure—Unsupervised Machine Learning-Based Approach"

_biomedicines, 2022, doi:10.3390/biomedicines10071514_

Round 1

Reviewer 1 Report

In the paper " Novel phenotyping for acute heart failure. Unsupervised machine learning-based approach" the authors analyze the natural phenotypic heterogeneity of the AHF population and evaluate possibilities offered by clustering (unsupervised machine learning technique) in medical data assessment. 

The research design is appropriate, the study is well written and very interesting for the readers 

Minor points

- Please add a flow chart of the study 

- Please add a figure in which the principal clinical, laboratoristic and echocardiographic features for every cluster are shown 

Reviewer 2 Report

This study is a retrospective analysis of 381 patients who were hospitalized due to acute heart failure. Those patients came from two AHF registries (one in 2010-2012, another one in 2016-2017). 63 clinical and laboratory parameters were used for analysis. K-medoids algorithm was implemented to create clusters and optimized based on the Davies-Bouldin index. They extracted six novel phenotypes of acute heart failure patients with different pathophysiology and prognosis. They thought these results can be valuable for future AHF trial constructions and more customized treatment. This paper is well written, but some viewpoints should be clarified.

# The study cohort consisted of two AHF registries separated for at least 5 years. Was there any difference in these two registries' study design and patients’ demographic data?

# This study used 63 clinical and laboratory variables for analysis. As we know, the management of heart failure improved a lot in recent 10 years. Many guideline-directed medical treatments were implanted such as Angiotensin receptor-neprilysin inhibitor (ARNI), Sodium-glucose co-transporter-2 (SGLT2) inhibitors, a new class of beta-blockers and mineralocorticoid receptor antagonists (MRAs). Could these medications have any impact on cluster analysis?

# How to use these proposed clustering to customize our acute heart failure management? Please discuss more and choose examples if possible.

Round 2

Reviewer 2 Report

You answer my questions.

Author Response

Thank you.